# Type I Interferon Signaling Controls Gammaherpesvirus Latency In Vivo

**DOI:** 10.3390/pathogens11121554

**Published:** 2022-12-17

**Authors:** Johannes Schwerk, Lucas Kemper, Kendra A. Bussey, Stefan Lienenklaus, Siegfried Weiss, Luka Čičin-Šain, Andrea Kröger, Ulrich Kalinke, Christopher M. Collins, Samuel H. Speck, Martin Messerle, Dagmar Wirth, Melanie M. Brinkmann, Hansjörg Hauser, Mario Köster

**Affiliations:** 1Model System for Infection and Immunity, Helmholtz Centre for Infection Research (HZI), 38124 Braunschweig, Germany; 2Institute of Genetics, Technische Universität Braunschweig, 38106 Braunschweig, Germany; 3Viral Immune Modulation Research Group, Helmholtz Centre for Infection Research (HZI), 38124 Braunschweig, Germany; 4Institute for Laboratory Animal Science, Hannover Medical School, 30625 Hannover, Germany; 5Institute of Immunology, Hannover Medical School, 30625 Hannover, Germany; 6Viral Immunology, Helmholtz Centre for Infection Research (HZI), 38124 Braunschweig, Germany; 7Innate Immunity and Infection, Helmholtz Centre for Infection Research (HZI), 38124 Braunschweig, Germany; 8Institute for Medical Microbiology, Otto von Guericke University Magdeburg, 39120 Magdeburg, Germany; 9Institute for Experimental Infection Research, Twincore, 30625 Hannover, Germany; 10Department of Microbiology & Immunology, Emory University Vaccine Center, Emory University School of Medicine, Atlanta, GA 30322, USA; 11Department of Virology, Hannover Medical School, 30625 Hannover, Germany

**Keywords:** murine gammaherpesvirus 68, latency, chronic infection, type I interferon

## Abstract

Gammaherpesviruses, such as Epstein-Barr virus and Kaposi’s sarcoma-associated herpesvirus, are important human pathogens involved in lymphoproliferative disorders and tumorigenesis. Herpesvirus infections are characterized by a biphasic cycle comprised of an acute phase with lytic replication and a latent state. Murine gammaherpesvirus 68 (MHV-68) is a well-established model for the study of lytic and latent life cycles in the mouse. We investigated the interplay between the type I interferon (IFN)-mediated innate immune response and MHV-68 latency using sensitive bioluminescent reporter mice. Adoptive transfer of latently infected splenocytes into type I IFN receptor-deficient mice led to a loss of latency control. This was revealed by robust viral propagation and dissemination of MHV-68, which coincided with type I IFN reporter induction. Despite MHV-68 latency control by IFN, the continuous low-level cell-to-cell transmission of MHV-68 was detected in the presence of IFN signaling, indicating that IFN cannot fully prevent viral dissemination during latency. Moreover, impaired type I IFN signaling in latently infected splenocytes increased the risk of virus reactivation, demonstrating that IFN directly controls MHV-68 latency in infected cells. Overall, our data show that locally constrained type I IFN responses control the cellular reservoir of latency, as well as the distribution of latent infection to potential new target cells.

## 1. Introduction

Gammaherpesviruses, including the family members Kaposi’s sarcoma-associated herpesvirus (KSHV) and Epstein Barr virus (EBV), are lymphotropic viruses that pose a significant risk to human health. Chronic gammaherpesvirus infection is associated with a number of malignancies, including Burkitt’s lymphoma, nasopharyngeal carcinoma, Kaposi’s sarcoma, primary effusion lymphoma, and multicentric Castleman’s disease [1,2]. Immunocompetent individuals control primary infection without the development of symptoms, eventually leading to the establishment of latent infection during which the viral genome is maintained as an episome [3,4]. Gammaherpesviruses like KSHV are able to reactivate from latency in immunocompromised individuals, leading to the production and dissemination of infectious virus particles.

Infection of mice with the murid herpesvirus 4 strain 68 (MHV-68) is an established small animal model to study host-pathogen interactions of gammaherpesviruses in vivo [5,6,7]. As observed for KSHV and EBV in humans, MHV-68 infection in immunodeficient mice is associated with the development of lymphomas [8]. Like all herpesvirus infections, the MHV-68 infection cycle is biphasic. Intranasal infection of MHV-68 in laboratory mice results in a productive infection in the nasal and respiratory mucosa [9]. The sequential expression of immediate-early, early, and late viral genes, including the replication and transcription activator (RTA, ORF50) and the tegument protein ORF52, characterize this lytic replication phase. The ORFs M3 and M9 (ORF65) are also transcribed abundantly during the lytic cycle [10]. The second phase of infection is chronic, known as latency, and is characterized by minimal expression of viral genes [11]. B cells are the predominant latency reservoir, followed by macrophages (especially in the peritoneum), splenic dendritic cells, and endothelial cells [12,13]. MHV-68 latency peaks around 16 days post-infection in secondary lymphoid tissues such as the spleen.

Control of MHV-68 latency in vivo has mainly been attributed to the adaptive immune mechanisms of T and B cells [14,15,16,17,18]. However, the current understanding of MHV-68 latency control suggests that concerted actions of intrinsic, innate, and adaptive immune response pathways are required to protect the host from reactivation-induced disease and the transformation of latently infected cells (reviewed in [19,20]). Interestingly, infection of NK cell-deficient mice via intranasal route suggested that NK cells are dispensable for control of lytic and latent MHV-68 infection [21,22]. Adler et al. suggested that during acute MHV-68 infection, the upregulation of CEACAM1 in the lung blocks NK-cell cytotoxicity [23]. However, another study indicated that NK cell depletion prior to MHV-68 infection by intrafootpad inoculation leads to increased viral titers in the popliteal lymph node [24]. An impact of type I IFN signaling on the reactivation capacity of MHV-68 has been demonstrated ex vivo in latently infected splenocytes and peritoneal macrophages [25]. However, the contribution of intrinsic type I IFN signaling in controlling gammaherpesvirus latency and upon in vivo reactivation is not well understood.

The IFN system is a key component of the innate immune response against viral infections [26]. Distinct classes of pattern recognition receptors (PRRs), such as Toll-like receptors (TLRs) or cytosolic RIG-I-like receptors (RLRs), detect invading pathogens and lead to the induction of IFN genes. Secretion of type I (mainly IFN-β and several IFN-α subtypes) and type III IFNs (IFN-λ) is a hallmark of the early innate immune response to viral infection [27]. Type I IFNs bind to the ubiquitously expressed heterodimeric type I IFN receptor (IFNAR), which leads to activation of the transcription factors STAT1 and STAT2 and induction of IFN-stimulated genes (ISGs), thus establishing an antiviral state. In addition, IFN activates effector cells of the innate and adaptive immune systems, thereby contributing indirectly to the immune response against viral infection.

Type I IFN receptor signaling is essential to control acute MHV-68 infection [25,28,29,30]. Furthermore, sustained expression of *Ifnb* mRNA during acute infection and early latency, as well as the counteraction of type I IFN gene induction by MHV-68 open reading frames 11, 36, and 64, has been demonstrated, highlighting the importance of type I IFN during these phases [31,32,33,34]. However, a suitable in vivo model to study the mechanism(s) of type I IFN controlling MHV-68 latency is still required. Here, we combined the detection of IFN activity by bioluminescent in vivo imaging with an adoptive transfer model of latently infected cells into *Ifnar1^−/−^* mice. We show that IFN expression and downstream signaling are induced in vivo during acute MHV-68 infection, as well as upon reactivation from latency. Using constitutive and inducible *Ifnar1^−/−^* mice, we identified an important contribution of type I IFN to MHV-68 latency control. Moreover, sequential splenocyte transfer experiments revealed that the transfer of latent infection to uninfected cells without detectable reactivation maintains MHV-68 latency, even in the presence of intact IFN signaling and innate as well as adaptive immune control.

## 2. Results

### 2.1. Primary MHV-68 Infection Induces an IFN Response In Vivo 

Primary MHV-68 infection in mice lacking the type I IFN receptor is fatal in our experimental conditions, which is in agreement with previous studies showing that type I IFN signaling is essential to control acute MHV-68 infection [25,28,29,30]. *Ifnar1^−/−^* mice succumbed to MHV-68 infection within 10 days, whereas WT mice remained asymptomatic and survived (Appendix A). To determine the spatio-temporal in vivo kinetics of IFN production during MHV-68 infection, we used an IFN-β reporter mouse expressing firefly luciferase under transcriptional control of the *Ifnb* promoter [35]. Mice were intranasally infected with MHV-68 and monitored for IFN-β reporter activity with whole-body in vivo imaging. We observed *Ifnb* induction on days seven and 14 in MHV-68-infected mice (Figure 1A). During that period, the constitutive activity of the *Ifnb* promoter in PBS-treated animals did not change. Luciferase reporter activity declined from day 14 to 21 in MHV-68-infected mice. The liver and salivary glands were prominent sources of IFN-β production. Due to the strong perfusion of the liver, IFN could be distributed to distal parts of the body via the bloodstream, thus acting on cells or tissues that were not infected with the virus. To test this, mice were sacrificed seven days after infection, and peripheral organs were assayed for induction of IFN responses and expression of viral ORF52 (a tegument protein involved in virion morphogenesis with lytic expression kinetics) [36]. ORF52 expression was detected in the lungs of infected mice but not in the spleen (Figure 1B). ISG expression, quantified by *Rsad2* mRNA expression, was significantly induced in the lungs and spleen seven days post-infection (Figure 1C). To determine IFN-mediated ISG induction in lymphoid organs, we established an adoptive transfer protocol for bioluminescence in vivo imaging. We used a reporter mouse expressing firefly luciferase under the control of the *Mx2* promoter (Mx2Luc), reflecting bona fide ISG expression [37]. Splenocytes were isolated from Mx2Luc mice and adoptively transferred into C57BL/6 albino mice. This experimental setup provided a highly sensitive tool to selectively visualize IFN activity in the spleen. Intravenous injection of 1 × 10^7^ Mx2Luc splenocytes led to the accumulation of cells in secondary lymphoid tissue (Figure 1D). Recipient mice were then infected intranasally with MHV-68 one day after adoptive splenocyte transfer, followed by whole-body in vivo imaging on different days after infection. Weak induction of *Mx2* was already detectable in the spleen one day after infection by luciferase reporter activity, which continuously increased over time (Figure 1D). These data document an IFN response in the spleen early after MHV-68 infection. The observed kinetics of *Mx2* induction indicate higher sensitivity compared to activation of the IFN-β reporter (compare Figure 1A). However, the lack of early detection of IFN-β expression can be explained by masking constitutive signals [35]. Furthermore, other types of IFN that induce *Mx2* expression, such as various IFN-α subtypes and IFN-λ, act cumulatively or might be preferentially induced early after infection [38,39]. On day seven post-infection, *Mx2* induction reached a level comparable to control mice that were stimulated by intravenous injection of 20 µg polyinosinic:polycytidylic acid (poly[I:C]) (Figure 1D). Poly[I:C] is a synthetic analog of double-stranded RNA (dsRNA), a molecular pattern associated with viral infections, and acts as a potent inducer of the IFN system. In vitro analysis of Mx2Luc reporter activity in the lungs and spleen of MHV-68 infected Mx2Luc reporter mice confirmed our in vivo imaging data (Figure 1E). Through the establishment of an adoptive Mx2Luc-reporter splenocyte transfer model, we uncovered IFN production already in the first days after infection. This early IFN acts on cells at uninfected, distal sites like the spleen, potentially protecting these organs from extensive viral replication.

### 2.2. IFN Signaling Is Required to Prevent MHV-68 Dissemination upon In Vivo Reactivation

In order to address if type I IFN signaling is required to control MHV-68 latency in vivo, we adoptively transferred latently infected CD45.1^+^ splenocytes into congenic (CD45.2^+^) WT or *Ifnar1^−/−^* recipient mice (Figure 2A). Adoptive transfer was performed at least 21 days after infection when splenic latency had become established [19]. This adoptive transfer model allowed for the transfer of latent infection via B cells into naïve recipient mice, as well as for donor and recipient cells to be distinguished from one another following adoptive transfer. Spleens of recipient mice were harvested 14 days after transfer, and CD45.2^+^ splenocytes were purified by cell sorting. RNA was isolated from CD45.2^+^ splenocytes of recipient mice and analyzed for lytic MHV-68 ORF52 expression. We detected elevated expression of viral ORF52 only in CD45.2^+^ splenocytes of *Ifnar1^−/−^* recipient mice but not in WT splenocytes (Figure 2B). Furthermore, *Ifnar1^−/−^* recipient mice exhibited symptoms of severe illness, and some eventually succumbed within 14 days after transfer. Wildtype recipient mice remained asymptomatic (data not shown). This suggests that intact IFN signaling is essential to restrict MHV-68 dissemination upon reactivation. Previously published data determined a reactivation rate of about 1 in 10^5^ cells for latently infected splenocytes in vitro [25]. To address if the in vivo reactivation rate is similar to our model, we performed a limiting dilution adoptive transfer of latently infected donor splenocytes into *Ifnar1^−/−^* recipient mice, which were then analyzed for MHV-68 reactivation in the spleen by detection of viral ORF52 expression 12 days after splenocyte transfer. The number of mice expressing viral ORF52 increased proportionally with the number of transferred latently infected splenocytes (Appendix A). One out of five mice transferred with 1 × 10^4^ donor cells, three out of five mice transferred with 5 × 10^4^ donor cells, three out of five mice transferred with 1 × 10^5^ donor cells, and all mice (4/4) transferred with 1 × 10^6^ donor cells were positive for viral ORF52 expression in the spleen (Appendix A). Hence, the in vivo reactivation rate from latently infected splenocytes is similarly low, as previously shown for ex vivo reactivation [25].

### 2.3. MHV-68 In Vivo Reactivation Induces an IFN Response 

To address if MHV-68 reactivation induces an IFN response, 1x10^7^ latently infected splenocytes from Mx2Luc reporter mice were adoptively transferred into WT or *Ifnar1^−/−^* mice. The transfer of wild-type splenocytes into *Ifnar1*-deficient mice allowed monitoring of ISG transcription in response to type I (mainly IFN-α/β) and type III (IFN-λ) IFNs, depending on cognate receptor expression. Recipient mice were then subjected to whole-body in vivo imaging at days three, five, and seven after adoptive transfer (Figure 2C). Mx2Luc reporter activity continuously increased in spleens and lymph nodes of *Ifnar1^−/−^* recipient mice indicating type I IFN secretion as a consequence of viral spread in these organs, whereas WT mice showed no reporter activity (Figure 2D). To further confirm that the observed *Mx2* induction in *Ifnar1^−/−^* mice is caused by loss of latency control and subsequent increase of virus burden, we used an MHV-68 H2bYFP reporter virus. This virus infects mice and establishes latency as efficiently as the MHV-68 WUMS used throughout this study [40]. Wild-type mice were infected with MHV-68 H2bYFP, and after the establishment of latency, splenocytes were adoptively transferred into WT or *Ifnar1^−/−^* recipient mice. Recipient mice were sacrificed 10 days after transfer, and splenocytes were analyzed by flow cytometry for YFP expression. *Ifnar1^−/−^* recipient mice showed a significantly increased frequency of YFP^+^ cells compared to WT mice (Figure 2E). Importantly, the frequency of YFP^+^ cells in *Ifnar1^−/−^* recipient mice was significantly increased over that of pooled donor splenocytes at the time of adoptive transfer (0.022%; dotted line). Similarly, MHV-68 genome copies were significantly higher in spleens of *Ifnar1^−/−^* recipient mice, as determined by qPCR analysis of viral gB DNA (Figure 2F). Virus titers were also significantly increased in spleens of *Ifnar1^−/−^* recipient mice (Figure 2G). The correlation of YFP^+^ cells or MHV-68 gB DNA with actual infectious viral titers validated the use of these two readouts as a measure for MHV-68 burden (Appendix A). These data show that a splenic environment incapable of type I IFN signaling results in loss of MHV-68 latency control and reactivation of the virus, accompanied by induction of IFN gene family members.

### 2.4. Type I IFN Signaling Does Not Prevent Low-Level Cell-to-Cell Transmission of MHV-68 during Latency 

While the absence of significant virus replication during latency has been reported [41], other studies suggested continuous subclinical low-level reactivation events during MHV-68 latency [15,42,43]. Our data show an increase of YFP^+^ splenocytes from day one to day 10 in recipient mice after the transfer of latently infected splenocytes (Figure 2E and Appendix A). Such an increase cannot be explained by the proliferation of latently infected B cells, instead proposing a model of dynamic latency where the transfer of MHV-68 to naïve cells occurs in the absence of detectable lytic replication. To investigate if virus transfer occurs during latency, we performed serial splenocyte transfer experiments (Figure 3A). Such serial adoptive splenocyte transfer addressed the potential transmission of latent MHV-68 from CD45.2^+^ donor cells to CD45.1^+^ recipient cells in a splenic environment capable of type I IFN receptor signaling. CD45.2^+^ mice were infected with MHV-68, and upon establishment of latency, splenocytes were adoptively transferred into congenic CD45.1^+^ recipient mice. After 10 days, CD45.1^+^ recipient splenocytes were isolated by cell sorting and transferred into *Ifnar1^−/−^* mice. This second round of adoptive transfer into *Ifnar1^−/−^* mice was performed to amplify and detect potential virus transmission from CD45.2^+^ donor cells to CD45.1^+^ recipient cells upon primary adoptive transfer. After 11 days, *Ifnar1^−/−^* recipient mice were sacrificed and analyzed for viral mRNA expression. Lytic ORF52 expression was detected in 7 of 8 *Ifnar1^−/−^* (87.5%) recipient mice (Figure 3B). Conventional RT-PCR for MHV-68 transcripts M3 and M9 further confirmed the ORF52 qPCR data (Figure 3C). Hence, low-level transfer of latent MHV-68 occurs in the absence of lytic replication despite intact type I IFN signaling. Re-analysis of the sorted CD45.1^+^ cell population confirmed purity of at least 99.8% (data not shown), showing that less than 1.5 × 10^4^ CD45.2^+^ donor cells are present in this population. Since the in vivo reactivation rate of latently infected splenocytes was determined to be comparable to previously published in vitro data (approximately 1 in 10^5^ cells) [25] (Appendix A), the chance of reactivation from 1.5 × 10^4^ contaminating CD45.2^+^ cells is less than 15%. As seven of the eight total recipient mice from the second serial transfer presented lytic transcription of MHV-68 ORF52 (Figure 3B), a major impact from potentially contaminating CD45.2^+^ cells can be excluded. Interestingly, the transfer of latently infected Mx2Luc splenocytes into WT recipient mice elicited a low IFN response (Figure 2C,D). *Mx2* induction was slightly elevated compared to mice that received splenocytes from uninfected donors. These data suggest that MHV-68 is transferred to previously uninfected splenocytes during latency, accompanied by the induction of low amounts of IFN.

### 2.5. Type I IFN Signaling Controls MHV-68 Latency by Acting on Latently Infected Splenocytes

Our data so far show the importance of type I IFN for the restriction of MHV-68 dissemination during latency. It is unclear, however, on which cells IFN is acting to control latent infection. IFN intrinsic control has previously been proposed and suggests that latently infected cells suppress MHV-68 reactivation through continuous, low-level exposure to tonic IFN. The authors showed that type I IFN receptor signaling facilitates the establishment of MHV-68 latency by exploiting the host’s IFN-induced IRF-2 expression ^33^. Furthermore, MHV-68 reactivates ex vivo with higher frequencies from *Ifnar1^−/−^* splenocytes and peritoneal macrophages than from WT cells [25]. To address the question of intrinsic latency control by type I IFN signaling, we generated an inducible *Ifnar1^−/−^* mouse (referred to as R26CreERT2 *x Ifnar1^fl/fl^*) by crossing conditional *Ifnar1^fl/fl^* mice to mice that constitutively express tamoxifen-inducible CreERT2 in the *Rosa26* locus [44,45]. While primary MHV-68 infection is fatal in conventional *Ifnar1^−/−^* mice (with different sensitivities depending on the mouse strain used), this novel inducible *Ifnar1^−/−^* mouse model allows the deletion of *Ifnar1* upon establishment of MHV-68 latency. To test the efficiency and control Cre-mediated recombination, a single dose of 2 mg tamoxifen was administered, and genomic recombination within the *Ifnar1* gene was evaluated in different organs by PCR two days later. Recombination was only observed in tamoxifen-treated mice, confirming tight control of the inducible Cre recombinase without noticeable basal activity (Appendix A).

R26CreERT2 *x Ifnar1^−/−^* mice were infected with MHV-68 H2bYFP, and after the establishment of latency (≥21 days after infection), 1 × 10^7^ splenocytes were adoptively transferred into conventional *Ifnar1^−/−^* recipient mice. These mice were subsequently treated with a single oral dose of 2 mg tamoxifen to induce *Ifnar1* deletion, specifically in the adoptively transferred splenocytes. Mice were sacrificed after seven days, and splenocytes were analyzed for MHV-68 DNA and viral YFP reporter expression (Figure 3D). MHV-68 gB DNA and frequency of YFP^+^ cells increased dramatically in three tamoxifen-treated *Ifnar1^−/−^* mice (Figure 3E,F, red dots). This observation was further supported by elevated titers of recoverable infectious virus (Figure 3G) and symptoms of severe illness in the same mice (data not shown). Such drastic manifestation of lytic MHV-68 infection upon reactivation was never observed that early (seven days after adoptive transfer) of WT splenocytes into *Ifnar1^−/−^* recipient mice. These observations suggest that type I IFN receptor signaling might have a direct, intrinsic effect on latently infected cells, limiting the extent of MHV-68 reactivation and subsequent virus dissemination from these cells.

## 3. Discussion

A comprehensive understanding of host-pathogen interactions and the underlying mechanisms of this interplay is essential for the development of new antiviral strategies. We set out to define the relevance of the type I IFN system during MHV-68 in vivo latency and to further identify cellular targets of IFN and determine the kinetics of IFN activity upon in vivo reactivation of MHV-68. For this, we used the Mx2Luc reporter mouse model. This model cumulatively senses minute amounts of type I and type III IFNs, thereby acting as a *bona fide* indicator of antiviral activity [37,46,47,48]. However, small amounts of basally expressed IFN mask complicate the detection of low-level de novo-induced IFN [35]. To overcome this limitation and further increase the sensitivity of the Mx2Luc reporter, we used an adoptive Mx2Luc splenocyte transfer model in combination with albino recipient mice. The negligible basal luciferase activity of transferred splenocytes in untreated recipient mice allowed for the detection of minimal stimulation of the Mx2Luc reporter. We took advantage of this experimental setup to monitor the kinetics of IFN induction and downstream signaling in lymphoid tissue during primary MHV-68 infection, as well as upon reactivation from latency. Susceptibility of *Ifnar1^−/−^* mice to primary MHV-68 infection poses a major obstacle when studying the role of type I IFN in MHV-68 latency in vivo. The adoptive splenocyte transfer model combined with different transgenic or mutant donor and recipient mice is a perfect tool to further delineate the contribution of type I IFN and its direct effect on latently infected cells and on surrounding uninfected cells. Using this approach, we were able to show that MHV-68 reactivation and subsequent dissemination occur at a significantly higher rate in an *Ifnar1^−/−^* environment. While a protective effect of type I IFN on potential virus target cells upon reactivation was expected, it is notable that it also acts directly on the latently infected cells. This led to a reduction in reactivation from this population. A previous study demonstrated transcriptional control of the late lytic MHV-68 M2 protein via IFN-induced IRF2, which leads to the suppression of lytic replication [33]. In theory, IFN-mediated control of the viral life cycle would result in a periodic reactivation-suppression pattern. Such a dynamic form of latency has been previously suggested for gammaherpesviruses [15,43] and is in line with our observations that MHV-68 is transmitted from latently infected splenocytes to naïve cells of recipient mice in the absence of detectable lytic reactivation. Such cell-to-cell transmission occurs in the presence of an active immune system and in WT recipient mice, in which cells were capable of producing and responding to type I IFNs. From this, we conclude that constitutive and even slightly increased levels of IFN are insufficient to inhibit low-level MHV-68 propagation.

Cell-to-cell transfer of MHV-68 in the absence of virion release into the intercellular space and de novo infection has been documented [41]. Importantly, the response to IFN within a population of cells is heterogeneous under sub-saturating IFN concentrations [46,49]. This could cause inefficient protection of potential MHV-68 target cells by IFN. Since IFN induction and downstream responses are barely detectable during latency, we hypothesize that latently infected cells are controlled, and uninfected cells are partially protected from de novo infection through low-level, basally expressed IFN [35,37,50] or anatomically restricted sites of IFN production. In line with this, tamoxifen-mediated *Ifnar1* deletion in latently infected donor cells had already led to substantially increased MHV-68 reactivation and dissemination in some of the recipient mice seven days after adoptive splenocyte transfer. Since the initial latent cell-to-cell transmission of MHV-68 is a stochastic process, not all mice in the treatment group presented signs of severe illness caused by strongly increased viral burden at the early time point of sacrifice (day seven after adoptive transfer). Salinas et al. observed that in CreERT2 transgenic mice, intraperitoneal injection of tamoxifen for five consecutive days during the latent phase of infection resulted in a slight (about 1.5-fold) increase in MHV-68 genome-positive cells compared to control mice [51]. This increase might be the result of off-target responses to the prodrug tamoxifen, which include lymphopenia and are mediated by ERα-unrelated, low-affinity effectors, particularly observed upon prolonged treatment [52,53]. In contrast to the protocol used by Salinas et al., we applied a single dose given by oral administration and observed a dramatic increase in viral genomes of more than 100-fold (Figure 3E) in three animals, while the remaining 14 animals were not affected by tamoxifen treatment. This suggests that the single application of tamoxifen avoids an unspecific increase in viral titers; rather, the controlled inactivation of *Ifnar1* promotes the dissemination of the virus in a fraction of animals.

Taken together, the adoptive splenocyte transfer model presented here provides an excellent tool to study the in vivo role of type I IFN signaling in maintaining MHV-68 latency and to assess the kinetics of IFN responses during acute infection, latency maintenance, and reactivation. Combination of highly sensitive IFN reporter mice together with constitutive and inducible *Ifnar1^−/−^* mice revealed that MHV-68 in vivo latency control is a dynamic process with permanent low-level reactivation events and silent cell-to-cell transmission of the virus that does not cause potent immune reactions to control these events. Furthermore, we uncovered that type I IFN signaling governs latency control in vivo by acting on both the latently infected cells as well as the uninfected potential virus target cells in their vicinity. Based on our findings, it is tempting to speculate that human gammaherpesvirus latency is also controlled by IFN. Future studies are required to investigate to which extent IFN also controls fulminant/induced lytic reactivation, which might hint toward IFN-based therapeutic interventions.

## 4. Methods

### 4.1. Mice and Ethics Statement 

All mice were of C57BL/6 genetic background and bred under SPF conditions at the Helmholtz Centre for Infection Research (Braunschweig, Germany). Mice were housed and handled in accordance with good animal practice as defined by FELASA. All mouse experiments were performed in compliance with the German Animal Welfare Law and were approved by the responsible state office (Lower Saxony State Office of Consumer Protection and Food Safety).

### 4.2. Viruses, Infection, and Tamoxifen Treatment of Mice

MHV-68 WUMS [7] and MHV-68 H2bYFP [40] were propagated on monolayers of M2-10B4 fibroblasts (ATCC CRL1972) and purified as described previously [54]. For intranasal infections, mice were anesthetized by intraperitoneal injection of a mixture of ketamine (100 µg/g body weight) and xylazine (5 µg/g body weight) and infected with 20 µL of virus diluted in PBS. Intravenous infections were performed in 100 µL PBS. Inducible *Ifnar1* knockout was induced by oral administration of a single dose of 2 mg tamoxifen (Ratiopharm) dissolved in 200 µL 20% ClinOleic emulsion (Baxter).

### 4.3. Adoptive Transfer and Sorting of Splenocytes 

Spleens were isolated from uninfected and MHV-68-infected donor mice upon establishment of latency (≥21 days post-infection). Control experiments confirmed latency at this stage and that the overall effects in recipient mice are not altered if spleens are taken between 21 and 42 days post-infection. Spleens were dissected and homogenized through a 70 µm cell strainer. Erythrocytes were removed by hypotonic lysis in Ammonium-Chloride-Potassium (ACK) buffer. Cells were stained with APC-conjugated anti-CD45.1 and FITC-conjugated anti-CD45.2 antibodies (BD Biosciences) at a 1:500 dilution. Cell sorting was performed using a FACS Aria cell sorter (BD Biosciences). Cells were washed in PBS, diluted to desired concentrations in a total volume of 100 µL PBS and adoptively transferred into recipient mice by intravenous injection.

### 4.4. Quantification of Luciferase Activity 

Ex vivo and in vivo quantification of luciferase reporter gene induction was performed as described previously [37]. Briefly, tissue samples were homogenized in proportional volumes of reporter lysis buffer (Promega) using the FastPrep system (MP Biomedicals). Lysates were assayed for luciferase activity in a reaction buffer (20 mM glycylglycine, 12 mM MgSO_4_, 1 mM ATP) containing luciferin (Synchem) using a single-tube luminometer (Berthold). For in vivo imaging, mice received 3 mg D-luciferin (Synchem) in 100 µL PBS intraperitoneally and were anesthetized using 2.5% isoflurane (Albrecht). Gray-scale images followed by bioluminescent images were acquired using an IVIS200 in vivo imaging system (PerkinElmer). Images were analyzed with Living Image 4.1 software. Luciferase activity was expressed as radiance [photons/sec/cm^2^/sr] and was illustrated by a rainbow scale as the intensity of radiance expressed as photons per second per cm^2^ per steradian (sr) [photons/sec/cm^2^/sr].

### 4.5. Quantification of Virus Burden from Tissue Samples

Total RNA was isolated from tissue samples by homogenization in TRIzol (Life Technologies) using the FastPrep system (MP Biomedicals). RNA was treated with DNase I (Promega) and reverse transcribed into cDNA using the Ready-To-Go You-Prime first-strand beads (GE Healthcare) and oligo(dT)_18_ primers. Genomic DNA was isolated from tissue samples using the QIAamp DNA Mini Kit (Qiagen). Amplification products were resolved by agarose gel electrophoresis to confirm the predicted size. qPCR reactions were performed using the QuantiTect SYBR green kit (Qiagen) in a Roche LightCycler 480 II (Roche Diagnostics). The specificity of the reaction was determined by melting curve analysis of the amplification products. Primer sequences used for PCR analysis: gB: forward 5′-CCTGGCTTTTATCGTGTTCG-3′, reverse 5′-TGACCTCCCTGACCCTCATA-3′; M3: forward 5′-CCAGGTCATTCACTGGTGTG-3′, reverse 5′-TGCTGGCATTCTGAAAGATG-3′; M9: forward 5′-TCCCTCCCTTTGAGGAAGAT-3′, reverse 5′-CCAAAATGATGGACCCTGTC-3′; ORF52: forward 5′-GGAGCAGTGGCTATGAGAAAA-3′, reverse 5′-TCTGTGTCTTGATTTGGACCTG-3′; *Rsad2*: forward 5′-GTCCTGTTTGGTGCCTGAAT-3′, reverse 5′-GCCACGCTTCAGAAACATCT-3′; *Actb*: forward 5′-ATTGTGATGGACTCCGGTGA-3′, reverse 5′-AGCTCATAGCTCTTCTCCAG; *Rps9*: forward 5′-CTGGACGAGGGCAAGATGAAGC-3′, reverse 5′-TGACGTTGGCGGATGAGCACA-3′. Flow cytometric analysis of the frequency of MHV-68 H2bYFP infected splenocytes was performed using a FACS Calibur (BD Biosciences). Recoverable infectious virus was determined based on a 50% tissue culture infective dose (TCID_50_) by infecting M2-10B4 cells with 10-fold serial dilutions of tissue homogenate [55].

### 4.6. Statistical Analyses

Individual experiments were performed two to four times. Figures display the results of one individual representative experiment. Figure 3E–G display cumulative data from two independent experiments. Data analysis was performed using Prism 7 (GraphPad Software); ns, *p* > 0.05; *, *p* ≤ 0.05; **, *p* ≤ 0.01; ***, *p* ≤ 0.001; ****, *p* ≤ 0.0001.

## Figures and Tables

**Figure 1 pathogens-11-01554-f001:**
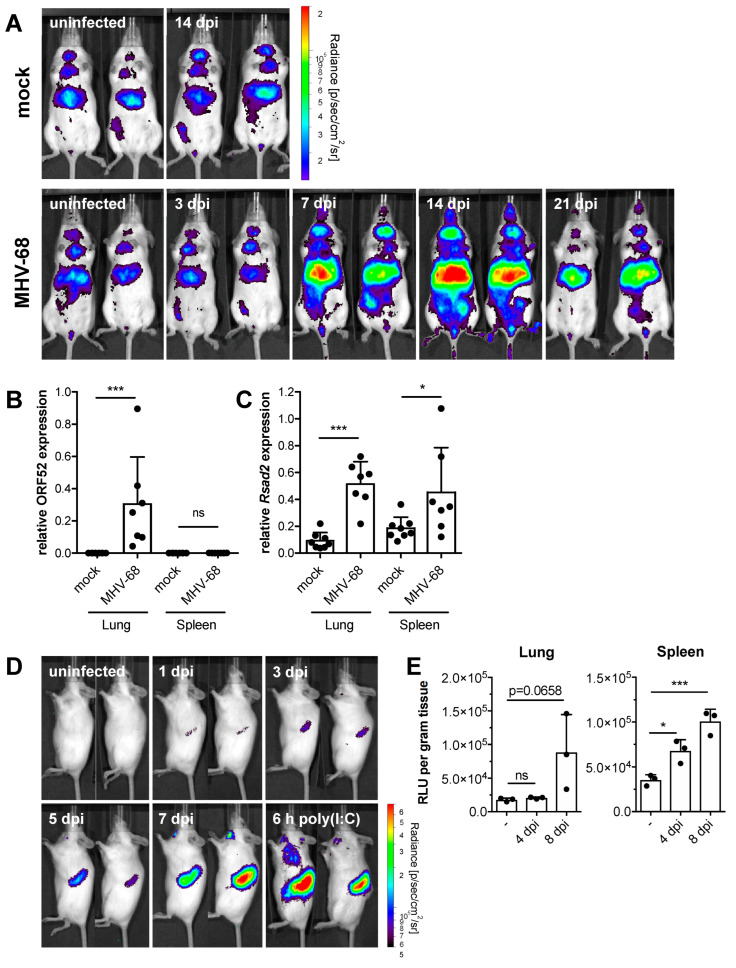
Acute MHV-68 infection induces IFN-β and ISG expression in vivo. (**A**) IFN-β luciferase reporter mice were intranasally infected with 5 × 10^4^ PFU MHV-68. Mock-infected control mice received PBS only. Whole-body in vivo imaging was performed at the indicated days post-infection. Two representative mice of each group are shown (n = 3–5). (**B**,**C**) Wild-type mice were intranasally infected with 5 × 10^4^ PFU MHV-68 and sacrificed seven days after infection. Relative expression of MHV-68 ORF52 and of mouse *Rsad2* in lung and spleen tissue was determined by qPCR (n = 7–8, mean ± SD). *p*-values were calculated by one-way ANOVA followed by Sidack’s Multiple Comparison Test. ns, *p* > 0.05; *, *p* ≤ 0.05; ***, *p* ≤ 0.001. (**D**) Whole-body in vivo imaging of luciferase activity of adoptively transferred Mx2Luc reporter splenocytes (1 × 10^7^ cells/mouse) upon intranasal infection of recipient mice with 5 × 10^4^ PFU MHV-68. Control mice were intravenously injected with 20 µg poly(I:C) and imaged after 6 h. Two mice from a representative experiment are shown. (**E**) In vitro analysis of luciferase activity in spleen and lung tissue four and eight days after intranasal infection of Mx2Luc reporter mice with 5 × 10^4^ PFU MHV-68 (n = 3, mean ± SD). *p* values were calculated by one-way ANOVA followed by Dunnett’s Multiple Comparison Test. ns, *p* > 0.05; *, *p* ≤ 0.05; ***, *p* ≤ 0.001.

**Figure 2 pathogens-11-01554-f002:**
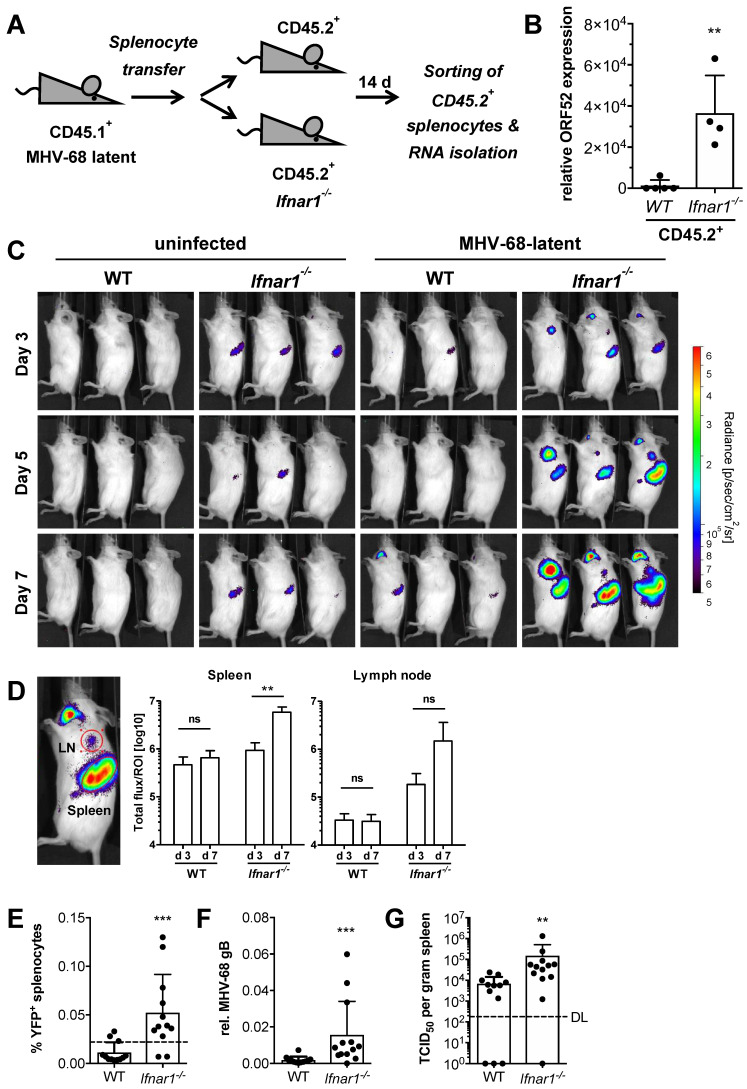
IFN signaling is required to prevent MHV-68 dissemination upon reactivation. (**A**) Schematic of the adoptive transfer model. CD45.1^+^ mice were intranasally infected with 5 × 10^4^ PFU MHV-68. After the establishment of latency (≥21 dpi), splenocytes were isolated from infected mice and adoptively transferred (1 × 10^7^ cells/mouse) into CD45.2^+^ WT or *Ifnar1^−/−^* recipient mice. (**B**) qPCR analysis of MHV-68 ORF52 expression in purified CD45.2^+^ splenocytes from WT and *Ifnar1^−/−^* recipient mice 14 days after adoptive transfer of latently infected splenocytes (n = 4–5, mean ± SD). *p*-value was calculated by unpaired *t*-test. **, *p* ≤ 0.01. (**C**) Whole-body in vivo imaging of luciferase activity upon adoptive transfer of 1 × 10^7^ splenocytes from latently infected Mx2Luc reporter mice into WT or *Ifnar1^−/−^* recipient mice. Control mice received uninfected splenocytes. (**D**) Quantification of luciferase activity by region of interest (ROI) analysis (red circles) of the spleen and brachial lymph node from WT and *Ifnar1^−/−^* recipient mice (n = 3, mean ± SD). *p*-values were calculated by two-way ANOVA followed by Bonferroni Multiple Comparison Test. ns, *p* > 0.05; **, *p* ≤ 0.01. (**E**) Frequency of YFP^+^ splenocytes in spleens of WT and *Ifnar1^−/−^* recipient mice 10 days after adoptive transfer of 1 × 10^7^ splenocytes from MHV-68 H2bYFP latent donor mice. The dotted line indicates the frequency (0.022%) of latently infected YFP^+^ donor splenocytes (pooled from 4 donor mice) immediately before adoptive transfer into WT and *Ifnar1^−/−^* recipient mice. (**F**,**G**) Relative quantification of MHV-68 genomic DNA (gB) by qPCR and recoverable infectious by TCID_50_ assay in spleen tissue of WT and *Ifnar1^−/−^* recipient mice 10 days after adoptive transfer of latently infected splenocytes (DL = detection limit). **, *p* ≤ 0.01; ***, *p* ≤ 0.001.

**Figure 3 pathogens-11-01554-f003:**
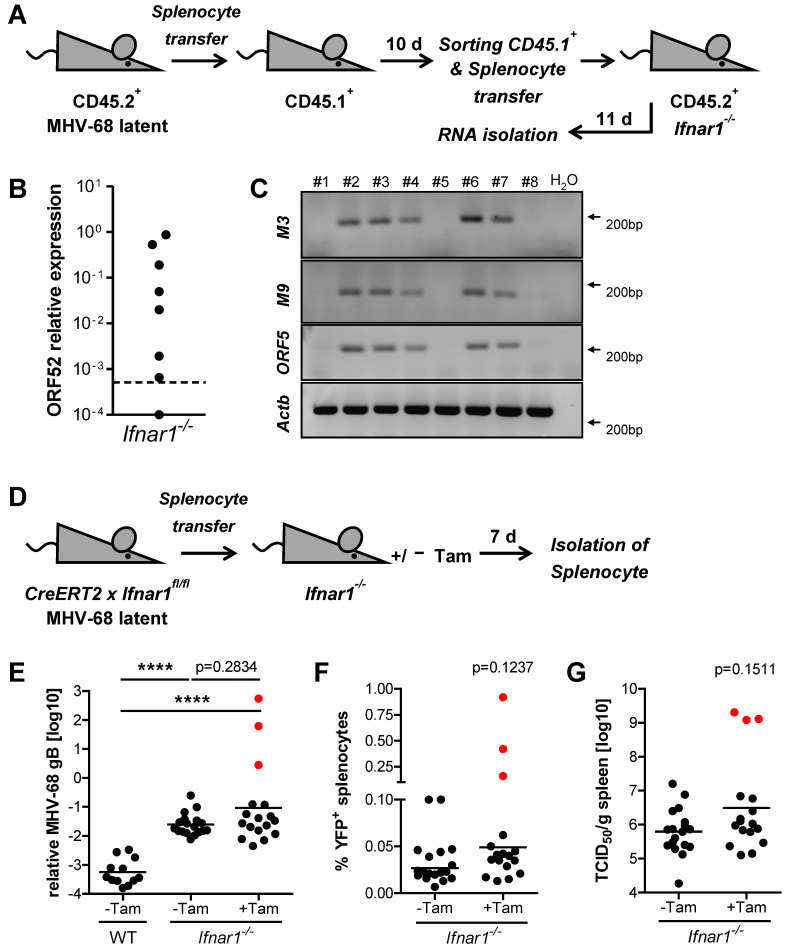
MHV-68 is transmitted during latency despite an intact type I IFN system. (**A**) Schematic of adoptive transfer model. Splenocytes were isolated and pooled from latently infected CD45.2^+^ mice (≥21 days after infection) and adoptively transferred (1 × 10^7^ cells/mouse) into CD45.1^+^ recipient mice (n = 8). Recipient mice were sacrificed after 10 days, and CD45.1^+^ splenocytes were purified by FACS and adoptively transferred (7.5 × 10^6^ cells/mouse) into *Ifnar1^−/−^* mice (n = 8). Total RNA was isolated from splenocytes 11 days after the transfer into *Ifnar1^−/−^* recipient mice. (**B**) Quantification of MHV-68 ORF52 expression by qPCR in splenocytes of *Ifnar1^−/−^* mice 11 days after adoptive transfer. The dotted line represents the level of relative ORF52 expression in pooled CD45.1^+^ donor splenocytes before adoptive transfer into *Ifnar1^−/−^* mice. (**C**) Conventional RT-PCR on MHV-68 transcripts M3, M9, and ORF52 in spleen tissue of *Ifnar1^−/−^* mice 11 days after the second adoptive transfer. (**D**) Schematic of adoptive transfer model. Splenocytes were isolated from latently infected (MHV-68 H2bYFP), tamoxifen-inducible *Ifnar1^−/−^* (R26CreERT2 × *Ifnar1^fl/fl^*) mice and adoptively transferred (1 × 10^7^ cells/mouse) into *Ifnar1^−/−^* recipient mice. Recipient mice were subsequently treated orally with (+Tam, n = 17) or without (-Tam, n = 18) a single dose of 2 mg tamoxifen. (**E**–**G**) Relative quantification of MHV-68 genomic DNA (gB) by qPCR, frequency of YFP^+^ splenocytes, and quantification of recoverable infectious by TCID_50_ assay in spleen tissue of recipient mice seven days after adoptive transfer (geometric mean is indicated). WT recipient mice received latently infected splenocytes in the absence of tamoxifen (n = 12). Red dots indicate measurements for the same three mice. *p*-values were calculated by Mann-Whitney’s U test. ****, *p* ≤ 0.0001.

## Data Availability

The data present in the study are available on request from the corresponding author.

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
