# Peer review of "Type I Interferon Signaling Controls Gammaherpesvirus Latency In Vivo"

_pathogens, 2022, doi:10.3390/pathogens11121554_

Round 1
Reviewer 1 Report
This is an interesting manuscript that uses the murine gamma-herpesvirus 68 (MHV68) model and very clever reporter and knockout mouse experiments to evaluate how type I interferon signaling influences and controls viral reactivation during the latent phase of infection. The results suggest that IFN signaling is induced by latent infection and functions to control sporadic reactivation and cell-to-cell spread, while also limiting disease. For the most part, the experiments are well-controlled and the results support the conclusions reached. I have a few concerns regarding imprecise or overstated interpretations and a suggestions regarding relevant publications that may have been overlooked by the authors. These are:
“eventually leading to establishment of immunologically silent latent infection” – note that latency is not immunologically silent as adaptive immunity is maintained and evolves over time during latent infection and is necessary to control long-term infection.
This statement is outdated “infection of NK cell-deficient mice suggested that NK cells are dispensable for control of lytic and latent MHV-68 infection”. Two more recent manuscripts highlight the roles of NK cells and type 1 IFN in MHV68 control – Lawler et al., JVI 2016 and 2020.
I realize that you show lethality with the doses used in this manuscript, but this is a bit of an overstatement – “MHV-68 infection in mice lacking the type I IFN receptor is fatal”. Lethality caused by MHV68 in the absence of type 1 IFN signaling is dependent on the viral dose and the mouse strain (129 background is susceptible with doses over 100 PFU inoculum, C57BL/6 is typically a much high dose that seems to vary by lab for some reason). See reference 16, as well as Paden et al., JVI 2010 for 129 data and Bennion et al., JVI 2019 for an example of C57BL/6 data. It should also be noted that latency was evaluated in reference 16, and also in Paden et al.
The experiment described in Fig. 3A, B, and C is interesting and agrees with the work of Freeman and colleagues, Ref. 10. However, this conclusion – “Hence, low-level immunologically silent transfer of latent MHV-68 occurs in absence of lytic replication and despite intact type I IFN signalling” was not definitively tested by the transfer experiment described for figure 3. Perhaps I am misunderstanding what is meant by “immunologically silent” (which is also used in the introduction and discussion), but I do not see where the immune response, which encompasses many different cell types and molecules, was tested here. Also, the data from Mx2-luc transfers in Figure 2 suggest that in vivo reactivation elicits IFN responses.
The tamoxifen-inducible deletion of IFNAR1-/- experiment is interesting and could be suggestive of a phenotype, but in a previous study with tamoxifen treatment of mice latently infected with MHV-68, Salinas and colleagues (PLOS Path., 2018) saw a slight increase in the number of latently infected cells in both WT and Cre-ERT2 C57BL/6 mice after tamoxifen treatment. Since a WT mouse control does not appear to have been performed in parallel for the current study, this caveat needs to be noted and discussed.
Reviewer 2 Report
This manuscript was submitted by Schwerk et al., which tried to demonstrate the importance of type I interferon signals in controlling gamma-herpesvirus MHV-68 in the mouse model. The results of this study are abundant and quite clear. Some questions still need to be addressed properly before this manuscript can be accepted for publication.
1. The background of MHV-68 needs to be introduced more, especially the viral proteins that appeared in this manuscript.
2. If the authors always used lytic protein ORF52 as a virus marker, how did you determine whether MHV-68 is in the latent or lytic phase in the mouse body? It is not clear how did the authors evaluate the latent rate or lytic rate of MHV-68 in the mouse body. Please explain it.
3. MHV-68 was detected dominantly in the lung and weakly in the spleen. Why did the authors choose splenocytes to perform adoptive transfer but not lung cells?
4. The locations between IFN activation and virus existence do not show a significant correlation.
Reviewer 3 Report
This is a very interesting study suggesting that Type 1 IFN plays a role in the control of gammaherpesvirus latency, although it appears that low-level transmission of MHV-68, a mouse model of gammaherpesvirus, still occurred despite induction of type 1 IFN. This suggests that type 1 IFN cannot fully prevent cell-to-cell viral transmission during latency, as the authors suggest; they also show convincing evidence that type 1 IFN is directly involved in the control of MHV-68 latency using a tamoxifen-inducible IFN knockout model. Overall, this is a very well-written paper, with appropriate citations, excellent figures and experimental design. Below are my minimal comments and suggestions:
1. First paragraph of results (page 2): “Mice were intranasally infected with MHV-68 and monitored for IFN-β reporter activity by whole-body in vivo imaging. We observed prolonged Ifnb induction, which was not yet detectable at day 3 and peaked between day 7 and 14 after infection (Figure 1A).”
a. Was prolonged ifnb induction observed in both infect and mock-infected mice? This language (“infection”) is ambiguous; please clarify if this was observed in both mock-infected and infected mice vs. just only infected mice in the body text. From the images you provided in Fig 1A, it appears that only mice infected w/ MHV-68 had detectable signals after day 7 post-infection.
2. Page 3: “On day 7 post infection, Mx2 induction reached a level comparable to control mice that were stimulated by intravenous injection of 20 μ g polyinosinic:polycytidylic acid (poly[I:C]) (Figure 1D).”
a. Please describe the function of poly [I:C] for the reader (i.e., how this serves as a valid positive control) either in the results, the methods, or in the introduction section. Probably most convenient if it were described in the results section.
3. Page 7: “These data show that a splenic environment incapable of type I IFN signalling results in loss of MHV-68 latency control and reactivation of the virus, which is accompanied by pro- nounced induction of IFN.”
a. If the splenic environment of the mouse is incapable of type 1 IFN signaling ("signaling" should only have 1 L, please fix this typo), the “pronounced induction” of IFN is presumably occurring outside of the spleen, correct? Are you referring to all IFNs, just type 1 IFNs, or other types of IFNs (e.g. type 3)? My understanding is that your MX2luc reporter shows bona fide transcription of interferon-stimulated genes, but it is unclear what the significance of “pronounced induction of IFN” in your language is: are you saying that transfer of latently infected splenocytes to KO mice resulted in detectable Mx2luc reporter activity, which should indicate that type 1 and type 3 IFNs are being produced but outside of the transferred spleen? Am I interpreting this correctly?
